# Synthesis and Anticancer Activity of Novel Dual Inhibitors of Human Protein Kinases CK2 and PIM-1 [note 1]

**DOI:** 10.3390/pharmaceutics15071991

**Published:** 2023-07-20

**Authors:** Patrycja Wińska, Monika Wielechowska, Mirosława Koronkiewicz, Paweł Borowiecki

**Affiliations:** 1Faculty of Chemistry, Warsaw University of Technology, 00-664 Warsaw, Poland; mwielechowska@ch.pw.edu.pl; 2Department of Biomedical Research, National Medicines Institute, 00-725 Warsaw, Poland; m.koronkiewicz@nil.gov.pl

**Keywords:** 4,5,6,7-tetrabromo-*N*,*N*-dimethyl-1*H*-benzimidazol-2-amine (DMAT) derivatives, protein kinase CK2, protein kinase PIM-1, dual protein kinase inhibitors, antitumor activity, apoptosis, autophagy

## Abstract

CK2 and PIM-1 are serine/threonine kinases involved in the regulation of many essential processes, such as proliferation, differentiation, and apoptosis. Inhibition of CK2 and PIM-1 kinase activity has been shown to significantly reduce the viability of cancer cells by inducing apoptosis. A series of novel amino alcohol derivatives of parental DMAT were designed and synthesized as potent dual CK2/PIM-1 inhibitors. Concomitantly with the inhibition studies toward recombinant CK2 and PIM-1, the influence of the obtained compounds on the viability of three human carcinoma cell lines, i.e., acute lymphoblastic leukemia (CCRF-CEM), human chronic myelogenous leukemia (K-562), and breast cancer (MCF-7), as well as non-cancerous cells (Vero), was evaluated using an MTT assay. Induction of apoptosis and cell cycle progression after treatment with the most active compound and a lead compound were studied by flow-cytometry-based assay. Additionally, autophagy induction in K-562 cells and intracellular inhibition of CK2 and PIM-1 in all the tested cell lines were evaluated by qualitative/quantitative fluorescence-based assay and Western blot method, respectively. Among the newly developed inhibitors, 1,1,1-trifluoro-3-[(4,5,6,7-tetrabromo-1*H*-benzimidazol-2-yl)amino]propan-2-ol demonstrates the highest selectivity and the most prominent proapoptotic properties towards the studied cancer cells, especially towards acute lymphoblastic leukemia, in addition to inducing autophagy in K-562 cells.

## 1. Introduction

Protein kinase CK2 (an acronym derived from the old misnomer “casein kinase 2”) and PIM-1 (proviral integration of Moloney virus-1) are protein serine/threonine kinases that have been implicated in cell growth and proliferation [1,2]. CK2 phosphorylates over 300 substrates and has multiple roles in the cell cycle, including in cell growth, proliferation, and survival [3,4]. The PIM kinase family contains three strongly evolutionary conservative isoforms: PIM-1, PIM-2, and PIM-3 [5]. Although CK2 and PIM kinases belong to different branches of the kinome [6], they exhibit structural and functional similarities. Unlike other eukaryotic kinases, both CK2 and PIM-1 demonstrate constitutive activity associated with a lack of phosphorylation sites in the activation loop [7,8]. Overexpression of these kinases occurs in many types of cancers, including leukemia, breast, and prostate cancer [9,10,11,12]. The increased catalytic activity of PIM-1 and CK2 kinases has been shown to enhance cell resistance to chemotherapy, inter alia, by inactivating BAD protein, a member of the Bcl-2 family [13], and interacting with the Myc factor [14]. It has been reported that the phosphorylation of BAD by CK2 and PIM kinases inhibits its proapoptotic functions in leukemia and prostate cancer, contributing to increased cell survival [13,15]. Moreover, PIM-1 overexpression combined with Myc leads to the development of the advanced form of prostate carcinoma [16]. The data show that overexpression of CK2 and PIM-1 kinases is a poor predictor in many cancers, such as prostate cancer and breast cancer, among others [17,18,19], and the reduction in CK2 and PIM-1 activity by chemical or molecular methods induces apoptosis in tumor cells [20,21,22].

Recent studies have shown that the contribution of CK2 and PIM-1 kinases to the regulation of transcription, differentiation, or signaling of DNA damage/repair systems is achieved by regulating survival pathways and hypoxia [23]. The activity of both CK2 and PIM-1 has been demonstrated to be elevated in hypoxia, which increases HIF-1 transcriptional activity (hypoxia-induced factor), whereas under normoxia, this factor is degraded [24]. In contrast, in hypoxia, its active form impacts the expression of many genes, stimulating angiogenesis and tumor cell resistance to chemotherapy [25,26,27]. Moreover, both kinases activate the transcriptional factor NF-κB, which is observed in transformed cells [28]. Furthermore, tumor transformation of lymphocytes with PIM-2 involvement depends on the activation of NF-κB [29]. In addition, PIM-1 is essential in activating the NF-κB pathway, which allows for the survival of prostate cancer cells treated with docetaxel, whereas PIM-1 knockdown or expression of a dominant negative protein sensitizes cells to the cytotoxic effects of docetaxel [30].

Over the last three decades, a plethora of potent and cell-permeable ATP-competitive inhibitors for CK2 kinase has been developed [4]. Among the most prominent examples are the polyhalogenated compounds depicted in Figure 1.

Essential chemical features necessary for efficient inhibitory activity toward CK2 have already been well-identified. Moreover, most of the structural characteristics of halogenated compounds were proven beneficial for the inhibition of the catalytic activity of PIM-1 [31,32]. Since complex signaling pathways and multiple targets are often involved in cancers, synergistically inhibiting multiple targets may be a more effective therapeutic strategy than a convenient single-target approach [33]. To date, quite a few multitarget CK2 inhibitors have been disclosed, which have the potential to become clinical candidates [31]. In this regard, the real breakthrough in the field of developing cell-permeable dual inhibitors of protein kinases CK2 and PIM-1 was achieved by discovering 1-(*β*-D-2′-deoxyribofuranosyl)-4,5,6,7-tetrabromo-1*H*-benzimidazole (TDB, also termed K164) [34,35]. These studies have shown that the use of dual inhibitors of CK2 and PIM-1 is beneficial for the reduction in cell proliferation and induction of apoptosis in cancer cell lines, i.e., cervical cancer (HeLa) and chronic myeloid leukemia (CML) [35,36]. The most recent studies demonstrated that inhibitor K164 is a promising compound that can be considered a potential active agent in targeted therapy in selected types of breast cancer [37].

In our previous studies [38,39], we evaluated the influence of the structure of the TBBi-alkanol side chain. We found that introducing hydroxyl groups into the aliphatic substituent attached to the hydrophobic TBBi scaffold is critical for the efficient inhibition of CK2. Although this structural fragment is located towards the outer part of the CK2 ATP-binding site, it generates additional polar interactions with a catalytic cavity, which increase the affinity toward the kinase receptor. In view of these findings, we also obtained a novel potent dual CK2/PIM-1 inhibitor, namely 3-(4,5,6,7-tetrabromo-2-methyl-1*H*-1,3-benzodiazol-1-yl)propan-1-ol (*N*^1^-PrOH-TBBi), which, after functionalization into its corresponding acetyl prodrug, turned out to be a very promising anticancer agent toward breast cancer cell lines [40]. Therefore, we expect that further structural modification of aliphatic substituents possessing efficient hydrogen bond-forming moieties installed at the appropriate distance from the TBBi core structure will enhance the affinity and selectivity of the designed 2-alkanol-TBBi derivatives toward titled kinases.

In this work, our ultimate goal was to design and synthesize novel amino alcohol derivatives of parental DMAT as potent dual CK2/PIM-1 inhibitors exhibiting promising anticancer activity. In this regard, intending to potentiate a DMAT-lead structure, we obtained a series of its model analogs differing in the structural topology of aliphatic amino alcohol substituents. Given that previous studies showed that breast cancer, as well as leukemic cells, is sensitive to dual CK2/PIM-1 inhibitor treatment [35,36,37], two leukemic cell lines, i.e., CCRF-CEM and K-562, and breast cancer cells, i.e., MCF-7, were used in the present study.

## 2. Materials and Methods

### 2.1. Chemistry

#### 2.1.1. General Procedure for the Synthesis of 2-Bromo-1*H*-benzimidazole (**2**)

Compound **2** was synthesized according to the procedure reported by Ellingboe et al. [41]. Br_2_ (12 mL, 0.24 mol) was added dropwise to a cooled (water bath, 5–10 °C) and mechanically stirred mixture of 2-mercapto-1*H*-benzimidazole (**1**, 10.0 g, 66.58 mmol), 48% aqueous HBr (10 mL), and glacial AcOH (100 mL) over 25 min. The mixture warmed slightly (40–45 °C) during the addition, and additional glacial AcOH (50 mL) was added to aid in the stirring of the thick mixture. After the addition was complete, stirring was continued at room temperature for 4 h. Afterwards, H_2_O (200 mL) was added, and the resulting solution was cooled in an ice bath (0–5 °C). The pH was adjusted to 4 with solid NaOH (ca. 40 g), and the precipitate was collected by filtration to afford a crude product, which was further purified by recrystallization from acetone to afford 2-bromo-1*H*-benzimidazole (**2**, 6.93 g, 35.17 mmol, 53% yield) as a white solid. Mp 194–196 °C (acetone) [41] 190–192 °C (acetone); *R*_f_ [CH_2_Cl_2_/MeOH (99:1, *v*/*v*)] 0.24; ^1^H NMR (500 MHz, DMSO-*d*_6_): *δ* 13.18 (br. s, 1H), 7.51 (s, 2H), 7.19 (m, 2H), 2.07 (s, 1H); ^13^C NMR (126 MHz, DMSO-*d*_6_): *δ* 139.7, 127.2, 122.2, 114.6; MS (ESI-TOF) *m/z*: [M + H]^+^ Calcd for C_7_H_6_BrN_2_^+^ *m/z*: 196.9709, Found 196.9697; FTMS (ESI-TOF) *m/z*: [M + H]^+^ Calcd for C_7_H_6_BrN_2_^+^ *m/z*: 196.97089 and 198.96884, Found 196.97125 and 198.96854; GC [200–260 (10 °C/min)]: *t_R_* = 3.16 min.

#### 2.1.2. General Procedure for the Synthesis of 2,4,5,6,7-Pentabromo-1*H*-benzimidazole (**3**)

Compound **3** was synthesized according to the procedure reported by Andrzejewska et al. [42]. Br_2_ (8 mL, 160 mmol) was added portionwise to a stirred and refluxed suspension of 2-bromo-1*H*-benzimidazole (**2**, 1.5 g, 7.6 mmol) in H_2_O (60 mL) within 6 h. The reflux was continued for 24 h under irradiation of purple LEDs (390 nm) (for details, see Appendix A). Afterward, the reaction mixture was cooled, and the orange precipitate was filtered off under suction. The collected solid was dissolved in MeOH/25% NH_3aq._ (80 mL, 3:1, *v*/*v*) and treated with charcoal (1.3 g) and Celite^®^ 545 (3.5 g). After filtering the solids under suction, the pale-yellow solution of the permeate was brought to pH 4–5 with conc. AcOH (150 mL), and the formed precipitate was recrystallized from a mixture of MeOH/H_2_O (50 mL, 1:1, *v*/*v*) to afford 2,4,5,6,7-pentabromo-1*H*-benzimidazole (**3**, 1.65 g, 3.22 mmol, 42%) as a yellowish solid.

Yield, 42% (1.65 g); yellowish solid; *R*_f_ [hexane/AcOEt (50:50, *v*/*v*)] 0.15; ^1^H NMR (500 MHz, DMSO-*d*_6_): *δ* Not found; ^13^C NMR (126 MHz, DMSO-*d*_6_ + 2 drops of 2M HCl_aq._ and registered for 16 h): *δ* 142.7, 138.9, 138.0, 131.3, 121.0, 120.9, 110.3; FTMS (ESI-TOF) *m/z*: [M + H]^+^ Calcd for C_7_H_2_Br_5_N_2_^+^ *m/z*: 512.60884, Found 512.60913.

#### 2.1.3. General Procedure for the Synthesis of Dual CK2/PIM-1 Inhibitors—TBBi Amino Alcohol Derivatives (**4**–**11**)

Compounds **4**–**11** were synthesized according to the procedure reported by Kazimierczuk and Pinna et al. [43].

*Method A:* A mixture of 2,4,5,6,7-pentabromo-1*H*-benzimidazole (**3**, 150 mg, 0.30 mmol) and the respective amino alcohol (4.9 equiv) in anhydrous EtOH (4.5 mL) was heated in an ace pressure tube (bushing type, back seal, *V* = 15 mL, L × O.D. 10.2 cm × 25.4 mm, Sigma Aldrich (Darmstadt, Germany): Z181064) at 110–115 °C for 72 h. Afterward, the reaction mixture was cooled to room temperature, and the volatiles were evaporated under reduced pressure using a rotavap. The oil residue was purified by a column chromatography using a sequential mixture of hexane/AcOEt (500 mL, 50:50 *v*/*v*) and CHCl_3_/MeOH (500 mL, 90:10 *v*/*v*) as an eluent to afford the desired solid-state products (**4**, *rac*-**6**, **8**, *rac*-**9**, **10**, and **11**).

*Method B:* A mixture of 2,4,5,6,7-pentabromo-1*H*-benzimidazole (**3**, 500 mg, 0.98 mmol) and racemic 2-hydroxypropylamine (1.10 g, 14.63 mmol, 1.13 mL) in anhydrous EtOH (7.5 mL) was heated in an ace pressure tube (bushing type, back seal, *V* = 15 mL, L × O.D. 10.2 cm × 25.4 mm, Sigma Aldrich (Darmstadt, Germany): Z181064) at 110–115 °C for 72 h. Afterward, the reaction mixture was cooled to room temperature, and the volatiles were evaporated under reduced pressure using a rotavap. The oil residue was purified by a column chromatography using a sequential mixture of hexane/AcOEt (500 mL, 50:50 *v*/*v*) and CHCl_3_/MeOH (500 mL, 90:10 *v*/*v*) as an eluent to afford the desired product, 1-[(4,5,6,7-tetrabromo-1*H*-benzimidazol-2-yl)amino]propan-2-ol (*rac*-**5**, 247 mg, 0.49 mmol, 50%), as a white solid.

*Method C:* A mixture of 2,4,5,6,7-pentabromo-1*H*-benzimidazole (**3**, 50 mg, 98 μmol) and optically active (*S*)- or (*R*)-1-aminopropan-2-ol (110 mg, 1.46 mmol, 113 μL) in dry PhCH_3_ (1 mL) was heated in an ace pressure tube (bushing type, back seal, *V* = 15 mL, L × O.D. 10.2 cm × 25.4 mm, Sigma Aldrich (Darmstadt, Germany): Z181064) at 100 °C for 48 h. Afterward, the reaction mixture was cooled to room temperature, and the volatiles were evaporated under reduced pressure using a rotavap. The oil residue was purified by a column chromatography using CHCl_3_/MeOH (95:5 *v*/*v*) as an eluent to afford the desired enantiomerically pure (2*S*)-1-[(4,5,6,7-tetrabromo-1*H*-benzimidazol-2-yl)amino]propan-2-ol [(*S*)-**5**, 46 mg, 91 μmol, 93% yield, >99% ee] or (2*R*)-1-[(4,5,6,7-tetrabromo-1*H*-benzimidazol-2-yl)amino]propan-2-ol [(*R*)-**5**, 42 mg, 83 μmol, 85% yield, >99% ee] as a white solid.

*4,5,6,7-Tetrabromo-N,N-dimethyl-1H-benzimidazol-2-amine (DMAT,* **4***)*

Synthesized according to *Method A* (Section 2.3). Yield, 29% (41 mg); white solid; *R*_f_ [CHCl_3_/MeOH (95:5 *v*/*v*)] 0.82; ^1^H NMR (500 MHz, DMSO-*d*_6_): *δ* 11.44 (br. s, 1H, N*H*), 3.13 (s, 6H, C*H*_3_); ^13^C NMR (126 MHz, DMSO-*d*_6_): *δ* 158.1 (*C*NCH_3_), 38.3 (*C*H_3_), the rest of the peaks were not detected; FTMS (ESI-TOF) *m/z*: [M + H]^+^ Calcd for C_9_H_8_Br_4_N_3_^+^ *m/z*: 477.74053, Found 477.74015.


*1-[(4,5,6,7-Tetrabromo-1H-benzimidazol-2-yl)amino]propan-2-ol (rac-*
**5**
*)*


Synthesized according to *Method B* (Section 2.3). Yield, 50% (247 mg); white solid; *R*_f_ [hexane/AcOEt (50:50, *v*/*v*)] 0.22 or *R*_f_ [CHCl_3_/MeOH (90:10, *v*/*v*)] 0.60 or *R*_f_ [CHCl_3_/MeOH (95:5 *v*/*v*)] 0.54; ^1^H NMR (500 MHz, acetone-*d*_6_): *δ* 11.04 (br. s, 1H), 6.48 (br. s, 1H), 4.70 (br. s, 1H), 4.07–3.92 (m, 1H), 3.56 (ddd, *J* = 13.6, 6.5, 3.5 Hz, 1H), 3.33 (ddd, *J* = 13.5, 7.3, 5.1 Hz, 1H), 1.19 (d, *J* = 6.3 Hz, 3H); ^1^H NMR (500 MHz, DMSO-*d*_6_): *δ* 11.41 (br. s, 1H), 6.62 (br. s, 1H), 4.95 (br. s, 1H), 3.83 (d, *J* = 5.9 Hz, 1H), 3.38 (ddd, *J* = 13.2, 6.5, 4.4 Hz, 1H), 3.26–3.18 (m, 1H), 1.10 (d, *J* = 6.2 Hz, 3H); ^13^C NMR (126 MHz, DMSO-*d*_6_ + 2 drops of 2M HCl_aq._ and registered for 16 h): *δ* 154.8, 153.2, 131.5, 130.5, 120.8, 118.0, 106.4, 103.7, 65.2, 50.1, 20.8; FTMS (ESI-TOF) *m/z*: [M + H]^+^ Calcd for C_10_H_10_Br_4_N_3_O^+^ *m/z*: 507.75110, Found 507.75086; HPLC [*n*-hexane-2-PrOH (95:5, *v*/*v*); f = 0.8 mL/min; λ = 225 nm; *T* = 25 °C (Chiralpak AD-H)]: *t_R_ =* 19.105 (*S*-isomer) and 21.524 min (*R*-isomer).


*1,1,1-Trifluoro-3-[(4,5,6,7-tetrabromo-1H-benzimidazol-2-yl)amino]propan-2-ol (rac-*
**6**
*)*


Synthesized according to *Method A* (Section 2.3). Yield, 28% (46 mg); beige solid; *R*_f_ [CHCl_3_/MeOH (95:5 *v*/*v*)] 0.55; ^1^H NMR (500 MHz, DMSO-*d*_6_): *δ* 11.69 (br. s, 1H), 6.76 (t, *J* = 5.9 Hz, 1H), 6.66 (br. s, 1H), 4.39–4.27 (m, 1H), 3.70 (ddd, *J* = 13.8, 6.1, 4.0 Hz, 1H), 3.49 (ddd, *J* = 13.9, 8.1, 5.8 Hz, 1H); ^19^F NMR (470 MHz, CD_3_CN): *δ* –77.26 (d, *J* = 7.4 Hz, 3F).; FTMS (ESI-TOF) *m/z*: [M + H]^+^ Calcd for C_10_H_7_Br_4_F_3_N_3_O^+^ *m/z*: 561.72283, Found 561.72251.


*3-[(4,5,6,7-Tetrabromo-1H-benzimidazol-2-yl)amino]propan-1-ol (*
**7**
*)*


Synthesized according to *Method A* (Section 2.3). Yield, 44% (67 mg); light green solid; *R*_f_ [CHCl_3_/MeOH (95:5 *v*/*v*)] 0.32; ^1^H NMR (500 MHz, DMSO-*d*_6_): *δ* 11.59 (br. s, 1H), 6.77 (t, *J* = 5.4 Hz, 1H), 3.50 (t, *J* = 6.1 Hz, 4H), 3.43 (q, *J* = 6.6 Hz, 2H), 1.71 (p, *J* = 6.5 Hz, 2H); FTMS (ESI-TOF) *m/z*: [M + H]^+^ Calcd for C_10_H_10_Br_4_N_3_O^+^ *m/z*: 507.75110, Found 507.75116.


*2-[Methyl(4,5,6,7-tetrabromo-1H-benzimidazol-2-yl)amino]ethanol (*
**8**
*)*


Synthesized according to *Method A* (Section 2.3). Yield, 37% (56 mg); white solid; *R*_f_ [CHCl_3_/MeOH (95:5 *v*/*v*)] 0.60; ^1^H NMR (500 MHz, DMSO-*d*_6_): *δ* 11.44 (br. s, 1H), 4.89 (br. s, 1H), 3.67–3.59 (m, 4H), 3.17 (s, 3H); FTMS (ESI-TOF) *m/z*: [M + H]^+^ Calcd for C_10_H_10_Br_4_N_3_O^+^ *m/z*: 507.75110, Found 507.75116.


*1-(Diethylamino)-3-[(4,5,6,7-tetrabromo-1H-benzimidazol-2-yl)amino]propan-2-ol (rac-*
**9**
*)*


Synthesized according to *Method A* (Section 2.3). Yield, 64% (112 mg); white solid; *R*_f_ [CHCl_3_/MeOH (95:5 *v*/*v*)] 0.48; ^1^H NMR (500 MHz, DMSO-*d*_6_): *δ* 8.22 (br. s, 1H), 5.32 (dt, *J* = 11.1, 5.5 Hz, 1H), 5.04 (ddd, *J* = 13.3, 5.8, 4.4 Hz, 1H), 4.85–4.76 (m, 1H), 4.24–3.80 (m, 12H); FTMS (ESI-TOF) *m/z*: [M + H]^+^ Calcd for C_14_H_19_Br_4_N_4_O^+^ *m/z*: 578.82459, Found 578.82454.


*2-[(4,5,6,7-Tetrabromo-1H-benzimidazol-2-yl)amino]propane-1,3-diol (*
**10**
*)*


Synthesized according to *Method A* (Section 2.3). Yield, 13% (20 mg); white solid; *R*_f_ [CHCl_3_/MeOH (95:5 *v*/*v*)] 0.10; ^1^H NMR (500 MHz, DMSO-*d*_6_): *δ* 6.39 (d, *J* = 7.7 Hz, 1H), 3.88–3.79 (m, 1H), 3.55 (qd, *J* = 10.7, 5.4 Hz, 4H); FTMS (ESI-TOF) *m/z*: [M + H]^+^ Calcd for C_10_H_10_Br_4_N_3_O_2_^+^ *m/z*: 523.74601, Found 523.74597.


*3-[(4,5,6,7-Tetrabromo-1H-benzimidazol-2-yl)amino]propane-1,2-diol (rac-*
**11**
*)*


Synthesized according to *Method A* (Section 2.3). Yield, 38% (60 mg); white solid; *R*_f_ [CHCl_3_/MeOH (95:5 *v*/*v*)] 0.13; ^1^H NMR (500 MHz, DMSO-*d*_6_): *δ* 6.61 (t, *J* = 5.3 Hz, 1H), 4.09 (d, *J* = 4.7 Hz, 1H), 3.70–3.60 (m, 1H), 3.51 (ddd, *J* = 13.4, 6.3, 4.6 Hz, 1H), 3.43–3.36 (m, 1H), 3.17 (d, *J* = 2.6 Hz, 2H); FTMS (ESI-TOF) *m/z*: [M + H]^+^ Calcd for C_10_H_10_Br_4_N_3_O_2_^+^ *m/z*: 523.74601, Found 523.74584.

### 2.2. Biological Evaluation

#### 2.2.1. Reagents and Antibodies

Dimethyl sulfoxide (DMSO), a molecular-biology-grade solvent used for all stocks of the chemical agents, was obtained from Carl Roth (Karlsruhe, Germany). All reagents used in flow cytometry analysis were purchased from BD Biosciences Pharmingen (San Diego, CA, USA). Information about antibodies is provided in Appendix A.

#### 2.2.2. Cloning, Expression, and Purification of Human CK2α, holoCK2, and PIM-1

CK2α, holoCK2, and PIM-1 were obtained according to Borowiecki [44] and Chojnacki [45]. The protein concentration in the final solution was 12.68 mg/mL for CK2α, 1.61 mg/mL for holoCK2, and 3.0 mg/mL for PIM-1 (determined by the Bradford method using bovine serum albumin as a standard) [46].

#### 2.2.3. Inhibition of Recombinant CK2 and PIM-1

The obtained compounds were tested for their inhibitory activity toward human CK2α, human CK2 holoenzyme, and PIM-1 using a P81 filter isotopic assay as described previously [39]. IC_50_ values were determined for the tested compounds at eight concentrations in the range of 0.005 to 400 µM. The experimental data were fitted to the sigmoidal dose–response equation, i.e., (variable slope) Y = Bottom + (Top-Bottom)/(1 + 10^((LogIC50−X)∗HillSlope)^, in GraphPad Prism (Prism 9, v. 9.0.1).

#### 2.2.4. Cell Culture and Agent Treatment

An acute lymphoblastic leukemia ALL cell line (named CCRF-CEM) was purchased from the European Collection of Authenticated Cell Cultures (ECACC), whereas MCF-7 (hormone-dependent breast adenocarcinoma), K-562 (human chronic myelogenous leukemia), and Vero cells (*Cercopithecus aethiops* kidney) were purchased from the American Type Culture Collection (ATCC, Manassas, VA, USA). For more details, see Appendix A.

#### 2.2.5. 3-(4,5-Dimethylthiazol-2-yl)-2,5-diphenyltetrazolium Bromide (MTT)-Based Viability Assay

After incubation with the tested compounds, an MTT test was performed as described previously [39]. Optical densities were measured at 570 nm using a BioTek microplate reader (BioTek Instruments, Inc., Winooski, VT, USA). All measurements were carried out in a minimum of three biological replicates.

#### 2.2.6. Detection of Apoptosis by Annexin V/propidium Iodide (PI) Labeling

MCF-7 cells were seeded in 6-well plates at 1.2 × 10^5^ cells/well, whereas CCRF-CEM and K-562 cells were seeded in 24-well plates at a density of 2 × 10^5^/mL. Cells were treated with the tested compounds used in 5 µM and 10 µM concentrations. Then, the plates were incubated for 48 h. After exposure to the examined compounds, the cells were collected and centrifugated at 200× *g* at 4 °C for 5 min, washed twice in cold phosphate-buffered saline (PBS), and subsequently suspended in binding buffer at 1 × 10^6^ cells/mL. Subsequently, 100-μL aliquots of the cell suspension were labeled according to the instructions of the respective manufacturer’s kit. Briefly, annexin V-fluorescein isothiocyanate and propidium iodide (BD Biosciences, Pharmingen, San Diego, CA, USA) were added to the cell suspension, and the mixture was vortexed, then incubated for 15 min at RT in the dark. A cold binding buffer (400 μL) was then added, and the cells were vortexed again and kept on ice. Flow cytometric measurements were performed within 1 h after labeling. Viable, necrotic, early, and late apoptotic cells were detected by flow cytometry using a BD FACSCanto II flow cytometer and analyzed using BD FACSDiva operating software (BD Biosciences, San Jose, CA, USA).

#### 2.2.7. Mitochondrial Membrane Potential (ΔΨm) Assay

Mitochondrial membrane potential was assessed by flow cytometry using 5,5′,6,6′-tetrachloro-1,1′,3,3′-tetraethylbenzimidazolocarbocyanine iodide (JC-1; Sigma-Aldrich, St. Louis, MO, USA). JC-1 undergoes potential-dependent accumulation in mitochondria. In healthy cells, the dye accumulates in mitochondria, forming aggregates with red fluorescence (FL-2 channel), whereas in dead and apoptotic cells, the dye remains in the cytoplasm in a monomeric form and emits green fluorescence (FL-1 channel). Cells were harvested by centrifugation 48 h post treatment, suspended in 1 mL of complete culture medium at approximately 1 × 10^6^ cells/mL, and incubated with 2.5 µL of JC-1 solution in DMSO (1 mg/mL) for 15 min at 37 °C in the dark. The stained cells were then washed with cold PBS, suspended in 400 µL of PBS, and examined by flow cytometry.

#### 2.2.8. Microscopic Examination

K-562 cells were seeded in 6-well plates at 2.5 × 10^5^/mL and subjected to the tested compounds. After 48 h of incubation, cells were centrifugated at 200× *g* at 4 °C for 5 min, subsequently suspended in PBS containing 2 µg/mL AO and Hoechst 33342, and incubated at 37 °C for 20 min in the dark. Subsequently, cells were centrifugated at 200× *g* at 4 °C for 5 min. Then, 5 µL of the sample was mounted on a glass slide and covered with a coverslip and examined under a Nikon ECLIPSE Y-TV55 fluorescent microscope.

#### 2.2.9. Fluorescence Intensity Assay

K-562 cells were seeded in 6-well plates at 2.5 × 10^5^/mL and subjected to the tested compounds. After 48 h of incubation, cells were centrifugated at 200× *g* at 4 °C for 5 min, subsequently suspended in PBS containing 2 µg/mL AO and Hoechst 33342, and incubated at 37 °C for 20 min in the dark. Subsequently, cells were centrifugated at 200× *g* at 4 °C for 5 min, and 150 µL of the sample was measured on dark plates (Sarstedt). The fluorescence intensity of AO at Ex. 502 nm/Em. 520–524 nm (aggregated or DNA complexed form), Ex. 457 nm/Em. 630–644 nm (aggregated or RNA complexed form), Ex. 540 nm/Em. 640–660 nm (red-stained lysosomes), and Ex. 488 nm/Em. 540–550 nm (yellowish stained lysosomes) and that of Hoechst (nuclei, Ex. 361 nm/Em. 497 nm) in K-562 cells was measured with a Synergy H4 Hybrid Multi-Mode Microplate Reader (BioTek Instruments, Inc., Winooski, VT, USA). The data were normalized to the intensity of Hoechst (nuclei, Ex. 361 nm/Em. 497 nm) in comparison to control cells serving as the reference point, showing 100%.

#### 2.2.10. Detection of Cell Cycle Progression by Flow Cytometry

MCF-7, CCRF-CEM, and K-562 cells were cultured in 6-well plates and treated with the tested compounds for 48 h. After exposure to the compounds, the cells were collected and washed with cold PBS and fixed at −20 °C in 70% ethanol for at least 24 h. Subsequently, cells were washed in PBS and stained with 50 μg/mL PI (propidium iodide) and 100 μg/mL RNase solution in PBS supplemented with 0.1% *v*/*v* Triton X-100 for 30 min in the dark at RT. Cellular DNA content was determined by flow cytometry employing a BD FACSCanto II flow cytometer (BD Biosciences, San Jose, CA, USA). The obtained DNA histograms were analyzed using MacCycle software (Phoenix Flow Systems, San Diego, CA, USA) for evaluation of the distribution of the cells in different phases of the cell cycle.

#### 2.2.11. Western Blotting

All the procedures are described in detail in Appendix A. The protein concentration was determined using a Bradford assay [46].

#### 2.2.12. Densitometry

For densitometry, immunoblots were scanned using G Box Chemi (Syngene, Cambridge, UK), and the density of each lane of phosphorylated and total protein was quantified using GeneSys software (Syngene, Cambridge, UK). Phosphorylated protein densities were normalized to GAPDH densities, assuming 1 for untreated cells; then, they were converted to a percentage of the appropriate control.

#### 2.2.13. Statistical Evaluation

Results are represented as mean ± s.e.m. of at least three independent experiments. Statistical analysis was performed using GraphPad Prism 5.0 software (GraphPad Software Inc., San Diego, CA, USA). Significance was determined using a one-way ANOVA analysis. The statistical significance of differences is indicated in figures by asterisks as follows: * *p* ≤ 0.05, ** *p* ≤ 0.01, and *** *p* ≤ 0.001.

### 2.3. Molecular Docking

#### 2.3.1. Molecular Docking Preparation

Molecular docking studies to establish favorable ligand binding geometries for both studied inhibitors, namely 1,1,1-trifluoro-3-[(4,5,6,7-tetrabromo-1*H*-benzimidazol-2-yl)amino]propan-2-ol (*rac*-**6**) and 3-[(4,5,6,7-tetrabromo-1*H*-benzimidazol-2-yl)amino]propane-1,2-diol (*rac*-**11**), were performed using AutoDock Vina v. 1.1.2 (http://autodock.scripps.edu/; accessed on 21 October 2021) [47]. First, ligands *rac*-**6** and *rac*-**11** in non-ionizable form were prepared with ChemAxon MarvinSketch v. 14.9.1.0 (http://www.chemaxon.com/marvin/; accessed on 9 September 2014). The initial geometries of the ligands with the minimum energy conformation (*E*_calc._ = −226.758 kJ/mol for *rac*-**6** and *E*_calc._ = −171.375 kJ/mol for *rac*-**11**) were optimized in Avogadro v. 1.2.0. (http://avogadro.cc/; accessed on 15 June 2016) using General Amber Force Field (GAFF) [48] and/or MMFF94 force field with 500 steps and the steepest descent algorithm. The visualization of the optimized geometries was performed using POV-Ray for Windows v. 3.7.0.msvc10.win64 licensed under the terms of the GNU Affero General Public License (AGPL3) (Figure 2). Afterward, the Gasteiger partial charges were calculated with AutoDock Tools v. 1.5.6 (ADT, S3 http://mgltools.scripps.edu/; accessed on 29 October 2022). In contrast, all torsion angles for each ligand were considered flexible, and all the possible rotatable bonds and non-polar hydrogens were determined. The final ‘ligand’ files were saved as PDBQT files (.pdbqt format) and were ready for the docking procedure disclosed below in Section 2.3.2. Molecular Docking Procedure.

The crystal structures of human protein kinases, namely CK2-α (PDB code: 4KWP) [35] of the highest available resolution (1.25 Å) and PIM-1 (PDB code: 4DTK) [49] with a resolution of 1.86 Å, were downloaded from the PDB database (http://www.rcsb.org/pdb/). The crude target proteins were prepared using the UCSF Chimera v. 1.11.2 package (http://www.cgl.ucsf.edu/chimera/; accessed on 2 December 2016) [50] after removing all nonstandard molecules, including 4,5,6,7-tetrabromo-1-(2-deoxy-beta-*D*-erythro-pentofuranosyl)-1*H*-benzimidazole (EXX), sulfate ion (SO_4_), 1,2-ethanediol (EDO), triethylene glycol (PGE), di(hydroxyethyl)ether (PEG), and dimethyl sulfoxide (DMS) in the case of 4KWP, as well as (5*Z*)-5-{2-[(3*R*)-3-aminopiperidin-1-yl]-3-(propan-2-yloxy)benzylidene}-1,3-thiazolidine-2,4-dione (7LI), EDO, and SO_4_ in the case of 4DTK. All the procedures are described in detail in Appendix A.

#### 2.3.2. Molecular Docking Procedure

Docking was performed using standard protocols as described in Appendix A. For validation of the docking calculations, two prominent kinase inhibitors crystalized with 4KWP and 4DTK, i.e., 4,5,6,7-tetrabromo-1-(2-deoxy-beta-*D*-erythro-pentofuranosyl)-1*H*-benzimidazole (EXX) for CK2-α and (5*Z*)-5-{2-[(3*R*)-3-aminopiperidin-1-yl]-3-(propan-2-yloxy)benzylidene}-1,3-thiazolidine-2,4-dione (7LI) for PIM-1, were docked as control ligands. The docking modes of each studied ligand (i.e., 1,1,1-trifluoro-3-[(4,5,6,7-tetrabromo-1*H*-benzimidazol-2-yl)amino]propan-2-ol (*rac*-**6**) and 3-[(4,5,6,7-tetrabromo-1*H*-benzimidazol-2-yl)amino]propane-1,2-diol (*rac*-**11**)) were clustered and ranked based on a mutual ligand–protein affinity expressed as absolute free binding energies (Δ*G*_calc_ (kcal/mol)), as well as the values of root mean square deviation (rmsd) in both modes with respect to the rmsd lower bound (l.b.) and the rmsd upper bound (u.b.). The rmsd values were computed with reference to the input structure submitted to docking simulations. For CK2-α (PDB code: 4KWP), the used random seed amounted to +1342461868 for *rac*-**6** and –2037069392 for *rac*-**11**, whereas for PIM-1 (PDB code: 4DTK), the used random seed amounted to +956047904 for *rac*-**6** and –769683352 for *rac*-**11**. The results of docking are collected in Appendix A. The optimized binding poses of *rac*-**6** and *rac*-**11** in hypothetical complexes with CK2-α and PIM-1 were visualized using PyMOL Molecular Graphics System software, v. 1.3, Schrödinger, LLC (https://www.pymol.org/; accessed on 13 October 2011).

## 3. Results

### 3.1. Chemical Synthesis

Taking DMAT (**4**) as the lead compound, a series of dual CK2/PIM-1 inhibitors were designed and synthesized in analogy to the methods already reported in the literature [41,42]. The general synthetic route is described in Figure 1. Briefly, in the first step, a commercially available 2-mercapto-1*H*-benzimidazole (**1**) was brominated with bromine (Br_2_) diluted in a mixture of 48% aqueous HBr and glacial acetic acid to afford 2-bromo-1*H*-benzimidazole (**2**) in 53% yield. Next, an exhaustive bromination of the resulting **2** with Br_2_ in boiling water performed under irradiation of purple LEDs (390 nm) for 24 h afforded the desired 2,4,5,6,7-pentabromo-1*H*-benzimidazole (**3**) in 42% yield. Finally, the aminolysis of the key intermediate **3** was carried out using the appropriate amino alcohol in anhydrous ethanol in a pressure glass tube reactor at elevated temperatures (110–115 °C) to afford DMAT (**4**) and its derivatives (**5**–**11**) in the yield range of 13–64%.

Due to the strong impact of the stereochemistry of xenobiotics on their biological activity in vivo, there is an urgent need to evaluate the single enantiomers of designed inhibitors toward target proteins, as well as cancer cell lines. Such an evaluation is incredibly valid especially when one of the optical isomers from the pair acts as the eutomer, while its counterpart behaves as the distomer. We envisioned that if the eudysmic ratio (ER) is high in the case of the developed chiral compounds (i.e., *rac*-**5**, *rac*-**6**, and *rac*-**11**), then the inhibitory activity of each enantiomer will significantly differ toward target kinases. Therefore, we found it pivotal for biological studies to elaborate on a highly efficient and stereoselective synthetic method for the preparation of both enantiomers of one of the chiral products (*rac*-**5)**. This task was accomplished by employing commercially available, optically pure (*S*)- and/or (*R*)-1-aminopropan-2-ol (>99% ee) as chiral building blocks. Unfortunately, conducting the reaction with (*S*)-1-aminopropan-2-ol using a standard protocol afforded (*S*)-**5** with only 39% ee. To our delight, a detailed screening of the reaction conditions, including the selection of organic solvent, as well as the evaluation of the effect of the reaction time and temperature on the stereochemical outcome, led to obtaining enantiomerically pure antipodes (*S*)-**5** and (*R*)-**5** (>99% ee) in the 85–93% yield range (for details, see Appendix A). Among the solvents used, only toluene (PhCH_3_) guaranteed that the enantiomers of **5** were isolated without undesired racemization. Other tested solvents (i.e., 1,4-dioxane, DMF, and CH_3_CN) achieved inferior results in terms of enantiomeric purity (26–83% ee). The considerable erosion in % ee-values with respect to the desired products ((*S*)-**5** and (*R*)-**5**) obtained from the reactions performed in aprotic polar solvents is interesting; however, the explanation for the racemization phenomenon is complex and definitely exceeds the scope of these studies.

All the resulting products (**4**–**11**) were characterized by single-proton nuclear magnetic resonance (^1^H-NMR) spectroscopy, high-resolution mass spectrometry (HR-MS), and high-performance liquid chromatography (HPLC). A lack of ^13^C-NMR spectra is typical for polyhalogenobenzimidazoles, since recording the narrow signals corresponding to quaternary carbon atoms is highly challenging for these compounds due to the electronic features of the TBB-ring possessing a N-H tautomeric proton. On the other hand, this problem is not observed in all the cases when the tautomeric proton is replaced by any substituent [45]. A series of modifications of the NMR experimental conditions implemented to overcome this drawback, including the significant extension of the time of analysis, changes in the values of relaxation times, application of high magnetic fields, and the use of lower temperatures to achieve slow exchange conditions, failed to obtain spectra with all the predicted signals. Interestingly, only the treatment of the samples with hydrochloric acid to avoid the existence of 1,3-tautomeric equilibrium by the protonation of nitrogen atom present in the imidazole ring allowed the ^13^C-NMR spectra to be recorded with the appropriate signals. Nevertheless, these results are subject to the risk of error, as no correction for the substituent effect was performed. For details, see copies of the recorded spectra appended in Appendix A.

### 3.2. Biological Evaluation

#### 3.2.1. Inhibition of Recombinant CK2 and PIM-1

Inhibition of the human CK2 catalytic subunit (CK2α), CK2 holoenzyme (CK2α2β2), and PIM-1 by the newly obtained compounds was evaluated using a radiometric assay (Table 1, Appendix A). The synthetic peptide RRRADDSDDDDD was used as the substrate of CK2, and peptide ARKRRRHPSGPPTA was used as the substrate of PIM-1. The values of the inhibition constant (*K*_i_) were calculated using the Cheng–Prusoff equation: *K*_i_ = IC_50_/(1 + [S]/*K*m) [51].

All the tested compounds were efficient inhibitors of both recombinant forms of CK2 and PIM-1, with *K*_i_ values in the range of 0.56–0.605 µM for CK2α2β2 and 0.052–0.267 µM for PIM-1. None of the newly obtained inhibitors inhibited CK2α2β2 or PIM-1 more strongly than the parent compound, DMAT (**4**); however, compounds **7**, **10**, and *rac*-**11** demonstrated lower *K*_i_ values for CK2α than for DMAT (**4**), i.e., 0.156 µM, 0.139 µM, and 0.151 µM, respectively. Among the newly obtained compounds, the *rac*-**11** derivative was the most efficient inhibitor of CK2α2β2, with *K*_i_ values equal to 0.089 µM, whereas *rac*-**5** was the most potent inhibitor of PIM-1, with *K*_i_ = 0.064 µM. The effectiveness of *rac*-**11** towards CK2 kinase can be attributed to the presence of two hydroxyl groups in its structure, which can additionally interact with structural water molecules accommodated in the enzyme’s active site. On the contrary, the weakest inhibitor of CK2, compound *rac*-**9** (*K*_i_ = 0.605 µM), has the largest substituent when compared to other studied inhibitors and can therefore potentially undergo clashes with amino acid residues present in the binding pocket of the enzyme.

#### 3.2.2. Cytotoxic Effect of DMAT Derivatives **4**–**11** toward Cancer Cell Lines: CCRF-CEM, K-562, MCF-7, and Non-Cancerous Vero Cells

To test the cytotoxicity of the DMAT derivatives, we treated the CCRF-CEM, K-562, MCF-7, and Vero cells with the newly synthesized compounds in the concentration range of 1.575–100 µM for 48 h. The representative plots demonstrating sigmoidal dose–response curves for compounds **4**–**11** are shown in Figure 3. The IC_50_ values describing the half-maximal effective concentration of each tested compound were calculated and are summarized in Table 2. Selectivity is an important feature of compounds demonstrating anticancer properties; therefore, the selectivity indices (SI) were calculated, as presented in Table 2. Using an MTT viability assay, we demonstrated that all tested compounds significantly decreased the viability of the studied cells, with IC_50_ values ranging from 9.66 µM to 41.53 µM. Among the newly obtained DMAT derivatives, *rac*-**6** and *rac*-**9** were the most cytotoxic toward the tested tumor cells (Table 2). The viability of CCRF-CEM and MCF-7 was most strongly reduced by *rac*-**6**, with IC_50_ values of 11.83 µM (for CCRF-CEM) and 9.66 µM (for MCF-7), whereas the viability of K-562 was the most strongly decreased by *rac*-**9**, with an IC_50_ value of 11.61 µM. Interestingly, *rac*-**6** demonstrated higher SI values than *rac*-**9**, with the highest value of 2.11 obtained for MCF-7. Considering the obtained results in terms of the lowest IC_50_ and the highest SI values, subsequent biological studies were devoted to *rac*-**6** and **4** (DMAT) as lead compounds.

#### 3.2.3. Induction of Apoptosis in CCRF-CEM, K-562, and MCF-7 Cells

In order to evaluate the proapoptotic properties of *rac*-6, we analyzed annexin V-binding to phosphatidylserine using flow cytometry. The results are shown in Figure 4 and Appendix A. The obtained results indicate that *rac*-6 induced apoptosis in CCRF-CEM at both concentrations, with the highest proportion of 47% of cells in early and late apoptosis after treatment with 10 µM conc. The examined racemate, *rac*-6, also induced apoptosis in MCF-7 cells at 10 µM conc. (statistically significant result), whereas K-562 cells were the least sensitive to *rac*-6 (no significant apoptosis observed). However, the percentage of total apoptotic K-562 cells was higher after treatment with 10 µM conc. of *rac*-6 than after treatment with parental compound 4 (DMAT), with values of 15.5% and 8.7%, respectively. We also observed the unusual presence of two different pools of unstained cells (considered alive) that were especially visible on cytograms obtained for K-562 cells treated with 10 µM conc. of *rac*-6 (Figure 4c). The results obtained for this line utilizing flow cytometry suggest a mechanism of death induction other than apoptosis for *rac*-6.

#### 3.2.4. Mitochondrial Membrane Potential (ΔΨm) in CCRF-CEM

Regarding the significant proapoptotic effect of *rac*-6 on CCRF-CEM cells, in the next step of our investigation, we measured mitochondrial membrane potential (ΔΨm) in leukemia cells treated with compounds 4 and *rac*-6 for 48 h (Figure 5). Incubation with the tested compounds caused mitochondrial membrane depolarization in a dose-dependent manner (as evidenced by the shift from a red to green fluorescence ratio). The most significant effect, i.e., 45% of cells with reduced ΔΨm, was obtained in cells treated with 10 µM conc. of *rac*-6. The results correlate well with annexin binding studies of *rac*-6-treated CCRF-CEM cells, suggesting the intrinsic nature of the apoptotic pathway.

#### 3.2.5. Detection of Autophagy in K-562 Cells

In order to conclude the mechanism of death of K-562 cells treated with the tested active agents, microscopic observations and a fluorescence intensity assay were performed after staining the cells with an acridine orange (AO). The employed dye is a cell-permeable fluorophore that can be protonated and trapped in acidic vesicular organelles (AVOs). Its metachromatic shift to red fluorescence is concentration-dependent, and therefore, acridine orange fluoresces red in AVOs, such as autolysosomes, can be detected quantitatively [52]. In addition, OA has the ability to intercalate between the nitrogenous bases of the nucleic acids present in cells and their labeling. DNA or RNA is digested during the autophagy process, and the resulting damage makes it easier to bind with OA molecules [52].

After 48 h of treatment with 4 and *rac*-6, K-562 cells, were subjected to staining with Hoechst (nuclei staining) and AO, followed by fluorescent microscopy and a fluorescence intensity measurement. As shown in Figure 6, control cells primarily displayed green fluorescence, with minimal red fluorescence, indicating a lack of acidic vesicular organelles (AVOs). Both 4- and *rac*-6-treated cells showed yellowish-stained lysosomes and a fold-increase in red fluorescent AVOs compared to the controls (Figure 6). Interestingly, cells treated with *rac*-6 demonstrated reduced green fluorescence (DNA and cytosol staining) in comparison to control cells (especially at higher concentrations of the compound). Since the obtained microscopic data were inconclusive, we additionally measured fluorescence intensity of AO at four different wavelengths, i.e., Ex. 502 nm/Em. 520–524 nm (aggregated or DNA complexed form), Ex. 457 nm/Em. 630–644 nm (aggregated or RNA complexed form), Ex. 540 nm/Em. 640–660 nm (red-stained lysosomes), and Ex. 488 nm/Em. 540–550 nm (yellowish stained lysosomes) (Figure 6b). The obtained quantitative data indicated that the fluorescence of K-562 cells treated with the tested compounds was increased in a dose-dependent manner for aggregated or DNA/RNA-complexed AO, as well as for yellowish-stained lysosomes, with the highest percentage of lysosomes reaching 324% of the control in a 15 µM conc. of the *rac*-6-treated cells.

Furthermore, the only statistically significant results were obtained for 15 µM conc. of the *rac*-6-treated cells. Moreover, the red fluorescence was reduced in the case of 10 µM conc. of the *rac*-6-treated cells. On the contrary, the red fluorescence was increased in 4-treated cells (both concentrations) and 15 µM conc. of the *rac*-6-treated cells up to 148% of the control, although the results in these cases were not statistically significant.

Because proton pump-driven lysosomal acidity generates a significant pH gradient, resulting in an efficient concentration of AO within the lysosome organelles and because the effectiveness of the AO concentration process is sufficient to create intralysosomal concentrations, leading to precipitation of the AO into aggregated granules, the obtained results indicate that some of the treated cells can, in fact, undergo autophagy, especially after treatment with *rac*-6. The results obtained for an aggregated form of AO correspond to the increase in the yellowish-stained lysosomes and red-stained lysosomes, with the exception of 10 µM conc. of the *rac*-6-treated cells.

#### 3.2.6. The Effect of 4 and *rac*-6 on Cell Cycle Progression in CCRF-CEM, K-562, and MCF-7 Cells

Since CK2 controls cell cycle progression and, consequently, its inhibition can affect the distribution of cells in the individual phases of the cell cycle, we tested the cell cycle progression in CCRF-CEM, K-562, and MCF-7 cells after treatment with 4 and *rac*-6. Representative plots DNA histograms with the calculations of cell percentages in each phase of the cell cycle are depicted in Figure 7. The results indicate differences in the mode of action of the two tested compounds, i.e., 4 led to S-phase arrest in all the tested cell lines, with a maximum of 57.7% of K-562 cells in that phase after treatment with 15 µM conc., whereas *rac*-6 induced G1-phase arrest in CCRF-CEM and MCF-7 cells, with a maximum accumulation of 49% of leukemia cells in that phase after using 10 µM conc. of this compound. Interestingly, *rac*-6 induced S-phase arrest only in K-562 cells, with a maximum of 56.1% of cells in that phase after 15 µM conc. of this compound. Furthermore, MCF-7 cells were observed in the sub-G1 phase (Figure 7d). Accumulation of cells in sub-G1 indicates DNA degradation, as often seen in apoptotic cells, confirming the results obtained with annexin V and PI staining.

#### 3.2.7. Intracellular Inhibition of Protein Kinase CK2 and PIM-1 in CCRF-CEM, K-562, and MCF-7 Cells

To confirm the intracellular inhibition of CK2 and PIM-1 by compounds 4 and *rac*-6 in the tested cell lines, we evaluated site-specific phosphorylation of Ser529 in NF-κBp65 (nuclear factor kappa-light-chain-enhancer of activated B-cells) and Ser112 in BAD after 48 h of treatment (Figure 8). The data obtained for p65 confirmed the intracellular inhibition of CK2 by the tested compounds in all the studied cell lines in a dose-dependent manner. Both tested compounds inhibited CK2-mediated phosphorylation of p65 to a similar extent, with the relative level of p65-P in the range of 0.18–0.47 µM in CCRF-CEM, 0.26–0.44 µM in K-562, and 0.19–0.73 µM in MCF-7. However, the most potent inhibition was observed in CCRF-CEM cells after treatment with 10 µM conc. of *rac*-6. Otherwise, Western blot data for BAD (a marker of PIM-1 activity) indicated inhibition of PIM-1 only in MCF-7 cells, with the most significant reduction in phosphorylated BAD in cells treated with 10 µM conc. of *rac*-6 (0.49 of control) (Figure 8b). Unexpectedly, the relative level of BAD-P in the drug-treated CCRF-CEM and K-562 cells was even higher than in control cells.

### 3.3. Molecular Docking

In order to rationalize the results of in vitro enzymatic assays and cellular activity in cancer cell lines of the most potent dual CK2-α/PIM-1 inhibitors with the highest selectivity factor (SI), comprehensive in silico enzyme–substrate docking calculations were performed. For this purpose, the respective TBBi derivatives (*rac*-6 and *rac*-11)were docked with receptor molecules prepared based on the crystal structures of both titled kinases, CK2-α (PDB code: 4KWP) [35] and PIM-1 (PDB code: 4DTK) [49], which were retrieved from the Protein Data Bank (PDB; https://www.rcsb.org/). As a result of molecular docking experiments, nine of the most energetically favorable binding modes for the ligand–protein complexes for ligands *rac*-6 and *rac*-11 and target CK2-α and PIM-1 proteins were generated. The results of their binding affinity energies expressed as Δ*G*_calc_ (kcal/mol) are presented in Appendix A). A visualization of the representative docking poses of *rac*-6 and *rac*-11 to CK2-α and PIM-1 with close contacts with amino acid residues located in the related ATP-binding sites of both studied kinases is presented in Figure 9.

Inspection of the productive pose of *rac*-6 in CK2-α (Figure 9A−C) showed that this inhibitor forms strong 2.1−2.2 Å-long hydrogen bonds with crystal waters (HOH-744 and HOH-954) present in the catalytic cavity. At the same time, the 4,5,6,7-tetrabromo-benzimidazole moiety of *rac*-6 is located deep inside the ATP-binding pocket, thus exhibiting strong hydrophobic interactions with Val53, Val66, and Val116. In contrast, *rac*-11 establishes hydrogen bonding between the backbone carbonyl oxygen of His160 residue and the hydrogen atom of the secondary hydroxyl moiety present in the propane-1,2-diol substituent (Figure 9D−F). Moreover, the CK2-α-*rac*-11 complex is stabilized through an additional water-mediated hydrogen bond between the imidazole-NH moiety and the oxygen atom of the bridging conserved water molecule (HOH-744). In this case, the ATP-binding pocket, which is surrounded by hydrophobic amino acids, was also occupied by the TBBi scaffold, providing alkyl CH–CH van der Waals (vdW) and π−alkyl interactions with Val53, Leu45, Val66, Glu114, Val116, Met163, and Ile174 residues. It is worth noting that docking simulations revealed that both TBBi inhibitors bind to CK2α, exploiting two different poses. In this context, *rac*-6 is anchored to the hinge region through Br1 and Br2 closer to Leu45, Val53, and Val66 residues (1.9−2.7 Å-long distance), while *rac*-11 is oriented through Br2 and Br3 closer to Glu114, Val116, Asn118, and Met163 residues (2.4−3.4 Å-long distance). As a consequence of the second orientation, ligand *rac*-11 can avoid unfavorable repulsion and steric clashes of the tailed hydrophilic propane-1,2-diol chain and hydrophobic residues located more profoundly in the ATP-binding pocket. The second plausible explanation for this orientation is that the additional primary hydroxyl group present in the amino alcohol substituent introduces a steric hindrance due to the proximity of Met163, which pushes *rac*-11 out of the hinge region. Interestingly, although *rac*-11 slightly protrudes from the ATP-binding site as compared to *rac*-6, the drop in the inhibitory potency of this compound was not detected during in vitro kinase activity assays.

In turn, complexes of PIM-1 and the selected top-scoring binding modes of *rac*-6 (Figure 9G−I) and *rac*-11 (Figure 9J−L) revealed that both inhibitors are accommodated inside the ATP-binding cleft with similar poses, explaining the marginal differences in inhibitory potency towards target protein in vitro. The only difference among the studied PIM-1 complexes concerns the feature of the H-bonding network between the inhibitors’ amino alcohol skeletons and interacting amino acid residues, as well as crystal water. The docking of *rac*-6 with PIM-1 protein shows that the secondary hydroxyl group of the alkyl substituent formed H-bond interactions with PIM-1 via amine groups of Lys67 or Asp186, whereas the TBBi-NH moiety interacted via crystal water (HOH-680). On the other hand, the secondary hydroxyl functionality of *rac*-11 interacted only with Lys67, while the TBBi-NH moiety was H-bounded with HOH-680. Interestingly, when comparing both inhibitors complexed with target kinases, one can see that in the case of CK2-α, it is the proton of imidazole-NH moiety that forms H-bonds with crystal water. In contrast, in the case of PIM-1, both inhibitors’ TBBi-NH scaffolds are engaged in the formation of H-bonding interaction with the crystal water.

Notably, no halogen bonding was detected among the docked ligand *rac*-6, suggesting that the fluorine atoms of the -CF_3_ group are less detrimental for polar interactions within CK2-α and PIM-1 receptor molecules. Therefore, it is more likely that the organofluorine moiety in the studied lead compound *rac*-6 is responsible for enhanced membrane permeation and higher bioavailability (due to the greater lipophilicity of bioisosteric fluorine compared to hydrogen), as well as improved metabolic stability (due to resistance of the C–F bond toward intracellular biotransformations compared to the C–H bond). Both aforementioned features can modulate the performance and pharmacokinetics of this compound in vivo but have no significant influence on the increased binding affinity of fluorinated derivatives to target kinases. Nevertheless, such a phenomenon concerning the physicochemical properties and pharmacological behavior of TBBi-based derivatives demands a deeper understanding.

## 4. Discussion

A series of novel amino alcohol derivatives (5–11) of parental DMAT (4) were designed and synthesized as potent dual CK2/PIM-1 inhibitors. All the compounds were obtained in up to 14% total yield after a three-step synthetic procedure, following a well-known route elaborated by Andrzejewska et al. [42], which includes: (i) bromination of 2-mercapto-1H-benzimidazole (1) with Br_2_ and 48% aqueous HBr in glacial AcOH, (ii) an exhaustive bromination of 2-bromo-1H-benzimidazole (2) with Br_2_ in boiling water and modified using additional LED (390 nm) irradiation to intensify the process, and (iii) aminolysis of the resulting key intermediate, namely 2,4,5,6,7-pentabromo-1H-benzimidazole (3), with commercial amino alcohols different than those used in the literature. Because the designed compounds included chiral small molecules, we also extended our synthetic efforts toward the preparation of the corresponding (R)- and (S)-enantiomers using commercially available chiral building blocks as starting materials. This strategy was applied to evaluate the structure–activity relationship with respect to the absolute configuration of stereogenic centers present in one of the studied DMAT analogs: (2S)- or (2R)-1-[(4,5,6,7-tetrabromo-1H-benzimidazol-2-yl)amino]propan-2-ol [(S)-5 or (R)-5].

Although the results of in vitro kinetics studies revealed that all inhibitors decrease the catalytic activity of recombinant CK2- and PIM-1 enzymes to a similar extent as a parental DMAT, the cytotoxicity and selectivity of the tested compounds were even better than in the case of DMAT. Moreover, the cytotoxic activity of the tested compounds corresponds better with their lipophilicity than their inhibitory activity. This correlation is in agreement with our previous findings demonstrating that the most promising compound characterized by a higher log*P* value can penetrate cell membranes more efficiently, therefore exhibiting improved intracellular inhibition of CK2 [40]. The present results were supported by molecular docking studies for the most selective compounds, i.e., *rac*-6 and *rac*-11. The in silico results demonstrated that both compounds are accommodated inside the ATP-binding cleft with similar poses, explaining the marginal differences in inhibitory potency towards target protein in vitro. These findings confirm that efficient inhibitors of recombinant CK2 should possess (nearby hydrophobic TBBi scaffold) a polar carboxyl or hydroxyl moiety, which are prone to forming strong hydrogen bonds similar to the phosphate groups present in the ATP physiological substrate.

Because cytotoxicity against tumor cells and selectivity are important features of compounds demonstrating anticancer properties, further investigations of biological action of the most promising compounds, i.e., *rac*-6 and 4, were performed. We observed that among tested cell lines, K-562, a Philadelphia (Ph) chromosome-positive (BCR-ABL-positive) leukemia cell line derived from chronic myeloid leukemia (CML) in blast crisis [53], was the most resistant toward the tested compounds. Previous studies demonstrated that CK2 expression was increased in leukemia cells from CML patients in blast crisis as compared to healthy peripheral blood mononuclear cells and showed that Bcr-Abl in K-562 cells physically interacts with CK2α, affecting its activity [54]. The therapeutic resistance of K-562 cells can also be partially correlated with an increased metabolic flux towards the Warburg phenotype, which promotes survival and proliferation [55]. It has been demonstrated that hexokinase-II (HK-II) is expressed predominantly in cancer cells, which promotes the Warburg metabolic phenotype and protects the cancer cells from drug-induced apoptosis. It was proven that K-562 cells have multifold higher levels of HK-II, glucose uptake, and endogenous ROS with respect to normal peripheral blood mononuclear cells [55].

The resistance of K-562 cells to the tested compounds correlates well with the lack of apoptosis in these cells after their treatment with *rac*-6 and 4. Moreover, the poor proapoptotic activity of the tested compounds towards K-562 cells also correlates with the lack of efficient intracellular inhibition of CK2 or PIM-1 protein kinases after 4- and *rac*-6-treatment. Considering that CK2 has been demonstrated as an essential mediator of BCR-ABL oncogenic signals and the BCR-ABL/CK2 complex is responsible for mediating BCR-ABL-induced cell proliferation and survival [56], the lack of efficient intracellular inhibition of CK2 by the tested compounds is in agreement with the poor proapoptotic activity of this treatment. We also observed that the phospho-BAD (Ser112) level in K-562 cells increased significantly after treatment with both inhibitors, which supports cell survival and prevents the occurrence of apoptosis. Moreover, it was demonstrated that the inactivation of BAD by PIM-1-mediated phosphorylation of Ser112 can affect BCL-2, an apoptotic cell death suppressor, which consequently leads to cell survival [57]. A similarly increased level of phospho-BAD (Ser112) was observed in TNBC MDA-MB-231 cells after treatment with a dual CK2/PIM-1 inhibitor, 1-(β-D-2′-deoxyribofuranosyl)-4,5,6,7-tetrabromo-1*H*-benzimidazole, named K164 (TDB) [37]. Interestingly, the same inhibitor reduced the catalytic activity of both CK2 and PIM-1 kinases, as manifested by decreased phosphorylated levels of Akt and BAD in K-562 cells, and induced apoptosis in this cell line [36].

Our present studies show that both tested compounds are prone to induce autophagy in the K-562 cell line. Although autophagy is a physiological cellular process that leads to the degradation and recycling of damaged cellular components [58], depending on the degree of activation, it can lead to death. In cancer, autophagy exhibits contradictory behavior, and depending on the cell type, it may be an important factor for the induction of cell death or tumor progression [59]. Previous studies have demonstrated that a well-established inhibitor of tyrosine kinase, imatinib, induces autophagy in the K-562 cell line and in primary cultures of patients with CML [60]. In turn, novel JAK inhibitor ruxolitinib (INCB018424) reported by Lin et al. [61] is able to notably decrease the expression of AKT, mTOR, and STAT autophagy inhibitor genes in K-562 cells relative to the control cell line. 

Among the tested cell lines, apoptosis was induced by *rac*-6 to the greatest extent in CCRF-CEM. Moreover, the proapoptotic activity of this compound was even better than the parental DMAT and corresponded to its strong ability to reduce ΔΨm, suggesting the intrinsic nature of the apoptotic pathway. The obtained results concerning CK2-mediated phosphorylation of NF-κB strongly correlate with the proapoptotic properties of *rac*-6, confirming the antiapoptotic role of this kinase. It was demonstrated that CK2 and its substrates protect cells from apoptosis by phosphorylating a wide range of proteins involved in the apoptotic response [62,63]. Interestingly, intracellular inhibition of CK2 in the *rac*-6-treated CCRF-CEM cells was not accompanied by inhibition of PIM-1 (increased level of phosphorylated BAD).

Among studied cell lines, the decreased level of (PIM-1)-mediated phosphorylation of BAD was observed only in MCF-7, and similarly to CCRF-CEM cells, CK2-mediated phosphorylation of p65 was reduced in MCF-7 cells treated with 4 and *rac*-6. The obtained results correlate with the ability of the tested compounds to induce apoptosis in breast cancer cells; however, its intensity is lower than in CCRF-CEM cells. The indicated differences in the proapoptotic efficacy of the tested inhibitors between CCRF-CEM and MCF-7 cell lines may be related to the caspase-3 deficiency in MCF-7 cells, which consequently undergo cell death due to a lack of typical apoptotic properties [64]. Interestingly, intracellular inhibition of CK2 after treatment with *rac*-6 detected in CCRF-CEM and MCF-7 cells correlates with G1-phase arrest in these cells, and it is opposite to S-phase arrest in K-562 cells. The G1-arrest that occurred in CCRF-CEM and MCF-7 cells after inhibition of CK2 is in agreement with the literature data, showing cell cycle regulation by CK2. It was demonstrated that treatment of cells with a selective inhibitor of CK2, i.e., CX-4945, resulted in reduced phosphorylation of a key cell cycle inhibitor protein [p21 (T145)] and increased the stability and levels of total p21 and p27 [65]. These cell cycle inhibitors are responsible for stopping the cell cycle in the G1 phase, activation of repair mechanisms, and apoptosis of cells with damaged DNA.

## 5. Conclusions

All the tested compounds were found to be efficient dual inhibitors of both recombinant forms of CK2 and PIM-1, with activity comparable to that of the parent compound. We concluded that the cytotoxic activity of the tested compounds corresponds better with their lipophilicity than their inhibitory activity. This correlation confirms our previous findings demonstrating that the most promising compound characterized by a higher log*P* value can penetrate cell membranes more efficiently.

Among the newly developed amino alcohol derivatives of DMAT, 1,1,1-trifluoro-3-[(4,5,6,7-tetrabromo-1*H*-benzimidazol-2-yl)amino]propan-2-ol (*rac*-6) demonstrates the most promising anticancer properties. Its ability to induce apoptosis in breast cancer cells can be attributed to efficient intracellular inhibition of both CK2 and PIM-1 activity. Moreover, *rac*-6 demonstrates the best anticancer properties towards acute lymphoblastic leukemia cells, inducing apoptosis via an intrinsic apoptotic pathway. Interestingly, both studied inhibitors, i.e., DMAT and *rac*-6, are able to induce autophagy in the BCR-ABL-positive chronic myeloid leukemia (CML) cell line.

The obtained results also support the concept that CK2 kinase is a vital factor for breast cancer cell survival and an appealing molecular drug target in the development of novel antineoplastic agents. Taking into account the obtained data, as well as our previous results demonstrating a synergistic effect of TBBi derivatives and classical cytostatics [66,67,68], it may be valuable to test a combination of *rac*-6 with 5-fluorouracil or methotrexate against breast cancer and leukemic cells.

## Data Availability

The data presented in this article are openly available.

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
