# Peer review of "Synthesis and Anticancer Activity of Novel Dual Inhibitors of Human Protein Kinases CK2 and PIM-1"

_pharmaceutics, 2023, doi:10.3390/pharmaceutics15071991_

Round 1

Reviewer 1 Report

After reading the manuscript "Synthesis and Anti-Cancer Activity of Novel Dual Inhibitors of Human Protein Kinases CK2α and PIM-1", I recommend that it be rejected as it stands.

Major

1. Authors should justify (and be sure to expand) the use of cell lines used in in vitro tests. The use of a vero line (not a human line) as a conditionally normal cell line is questionable.

2. The objects of study contain an enormous amount of bromine in their composition. This inevitably brings to mind potential problems with the solubility of bromobenzimidazoles. It is desirable to determine the kinetic or thermodynamic solubility of these compounds.

3. Based on the cytotoxicity data presented, these compounds cannot be called highly potent on cells (despite their nanomolar kinase activity). I recommend that the authors consider the study of the combination of classical cytostatics with the proposed inhibitors as an addition to this work.

language level normal

Author Response

Overview of Responses to Reviewer's #1 Comments:

COMMENT 1: Reviewer #1 stated the following: Authors should justify (and be sure to expand) the use of cell lines used in in vitro tests. The use of a vero line (not a human line) as a conditionally normal cell line is questionable.

RESPONSE 1: In order to justify the usage of the cancer cell lines, we added the following sentence in the introduction section: "Regarding that the previous studies showed that breast cancer, as well as leukemic cells, are sensitive to dual CK2/PIM inhibitors treatment [36‒38], two leukemic cell lines, i.e., CCRF-CEM and K-562 and breast cancer cells, MCF-7 were used in the present study."

To the best of our knowledge, the Vero cell line is a well-established model of normal cells, and many papers report on cytotoxicity using these cells. The list of a few of them is as follows:

  • Buachi C, Thammachai C, Tighe BJ, Topham PD, Molloy R, Punyamoonwongsa P. Encapsulation of propolis extracts in aqueous formulations by using nanovesicles of lipid and poly(styrene-alt-maleic acid). Artif Cells Nanomed Biotechnol. 2023 Dec;51(1):192-204. doi: 10.1080/21691401.2023.2198570. PMID: 37052886
  • Huang YJ, Zhong XL, Zang YP, Yang MH, Lin J, Chen WM. 3-Hydroxy-pyridin-4(1H)-ones as siderophores mediated delivery of isobavachalcone enhances antibacterial activity against pathogenic Pseudomonas aeruginosa. Eur J Med Chem. 2023 Sep 5;257:115454. doi: 10.1016/j.ejmech.2023.115454. Epub 2023 May 16. PMID: 37210837
  • Yang H, You M, Shu X, Zhen J, Zhu M, Fu T, Zhang Y, Jiang X, Zhang L, Xu Y, Zhang Y, Su H, Zhang Q, Shen J.Eur. Design, synthesis and biological evaluation of peptidomimetic benzothiazolyl ketones as 3CLpro inhibitors against SARS-CoV-2. J Med Chem. 2023 Sep 5;257:115512. doi: 10.1016/j.ejmech.2023.115512. Epub 2023 May 23.PMID: 37253309 
  • Kessler JC, Vieira V, Martins IM, Manrique YA, Ferreira P, Calhelha RC, Afonso A, Barros L, Rodrigues AE, Dias MM. The potential of almonds, hazelnuts, and walnuts SFE-CO2 extracts as sources of bread flavouring ingredients. Food Chem. 2023 Aug 15;417:135845. doi: 10.1016/j.foodchem.2023.135845. Epub 2023 Mar 11. PMID: 36924720
  • Staniszewska M, Kuryk Ł, Gryciuk A, Kawalec J, Rogalska M, Baran J, Kowalkowska A. The Antifungal Action Mode of N-Phenacyldibromobenzimidazoles. Molecules. 2021 Sep 8;26(18):5463. doi: 10.3390/molecules26185463.

COMMENT 2: Reviewer #1 stated the following: The objects of study contain an enormous amount of bromine in their composition. This inevitably brings to mind potential problems with the solubility of bromobenzimidazoles. It is desirable to determine the kinetic or thermodynamic solubility of these compounds.

RESPONSE 2: We agree with the reviewer's opinion that the studied compounds exhibit poor-to-moderate solubility in an aqueous biological media. Nevertheless, the pharmacological properties of druglike molecules are a more complex issue than only solubility in water. Therefore, we decided to implement a set of valuable parameters (i.e., physicochemical properties, lipophilicity, water solubility, pharmacokinetics, and drug-likeness) into the Supporting Information file, which might shed more light on the designed active agents in terms of "medicinal chemistry friendliness". In this regard, please see the section "4. Evaluation of ADMET profile for the obtained compounds" (Pages: S12-S16) in the Supporting Information file. These data include absorption, distribution, metabolism, excretion, and toxicity (ADMET) profiles for the synthesized compounds. The ADMET profile was evaluated using the SwissADME web-platform (http://www.swissadme.ch/) with a web-based interface. This online tool, provided by the Molecular Modeling Group of the Swiss Institute of Bioinformatics (SIB), is free of charge [please also see the respective reference: Daina, A.; Blatter, M.-C.; Baillie Gerritsen, V.; Palagi, P. M.; Marek, D.; Xenarios, I.; Schwede, T.; Michielin, O.; Zoete, V. Drug Design Workshop: A Web-Based Educational Tool to Introduce Computer-Aided Drug Design to the General Public. J. Chem. Educ. 2017, 94, 335–344, doi:10.1021/acs.jchemed.6b00596.].

Of course, according to the reviewer's request, the water solubility of the tested compounds was also determined using three different algorithms supported by the SwissADME tool - log S (ESOL), log S (Ali), and log S (SILICOS-IT), respectively.

We hope that the computed physicochemical descriptors, as well as predicted ADME parameters, pharmacokinetic properties, druglike nature, and medicinal chemistry friendliness of the small molecules reported in our paper, will support drug discovery in the future and that the reviewer will also find this information valuable, thus allowing us to publish our manuscript in Pharmaceutics journal. 

COMMENT 3: Reviewer #1 stated the following: Based on the cytotoxicity data presented, these compounds cannot be called highly potent on cells (despite their nanomolar kinase activity). I recommend that the authors consider the study of the combination of classical cytostatics with the proposed inhibitors as an addition to this work.

RESPONSE 3: The cytotoxicity of the tested compounds is different, but the most promising compound, i.e., rac-6 demonstrates relatively low IC50 values, especially for CCRF-CEM (11.83 µM) and MCF-7 (9.66 µM) after 48 h of incubation. This value is similar to IC50 determined for CX-4945, which is the first competitive inhibitor of CK2 approved for use in cancer therapy.

We agree with the reviewer's opinion that combination studies with classical cytostatics would be valuable for a better understanding of the active agents' mechanism of action and cytotoxicity potency. Although we find this issue very interesting, we plan to investigate the synergistic effects of such combinations in due course.

To underline this suggestion, we added the following sentence in the conclusion section:

"Taking into account the obtained data as well as our previous results demonstrating a synergistic effect of TBBi derivatives and classical cytostatics [67‒69], it could be valuable to test a combination of rac-6 with 5-fluorouracil or methotrexate against breast cancer and leukemic cells."

Reviewer 2 Report

This study designed a series of novel amino alcohol derivatives which exhibited significant effectiveness to inhibit the growth of tumor cells. This research had the potential to promote the therapy of tumors. There are still some suggestions for the authors to consider.

1. Some statistical results should be added to the result part of your paper.

2. The proportion of cells in Flow Cytometry should be exhibited in the primary figures.

The quality of the English language is enough for the readers to understand.

Author Response

Overview of Responses to Reviewer's #2 Comments:

COMMENT 1: Reviewer #2 stated the following: Some statistical results should be added to the result part of your paper.

RESPONSE 1: We agree with the reviewer's opinion and have added the statistical analysis of quantitative data of cell cycle analysis (Figure 7). The western blot results are not quantitative data but rather qualitative; therefore, they are not subject to statistical analysis.

COMMENT 2: Reviewer #2 stated the following: The proportion of cells in Flow Cytometry should be exhibited in the primary figures.

RESPONSE 2: The proportions of the cells are presented in bar graphs with statistical analysis, making them more readable (please see Figure 4a). We believe adding numbers to the original charts will result in a lack of readability (Figure 4c). Therefore, we would like to rebut this request to retain clarity. However, to avoid misleading, a tabular summary was attached to the Supplementary Information (see Table S2).

Reviewer 3 Report

This manuscript aimed to design and synthesize novel amino alcohol derivatives of parental DMAT as potent dual CK2/PIM-1 inhibitors by changing the substituents on N atom of DMAT. And also biological evaluation at molecular level and celluar level were performed to identify rac-6 as promising anti-cancer compound. Binding modes of Rac-6 was elucidated based on moelcular docking results. I would like to recommend this manuscript to be published on Molecules unless authors make the following revision.

 As for the results of molecular docking, owing to the similar inhibitory activity of rac-6 and DMAT, it will be better for authors to compare the binding modes between rac-6 and DMAT and its deviates with CK2, which will provide a deep insight of binding modes for readers. 

Author Response

Overview of Responses to Reviewer's #3 Comments:

COMMENT 1: Reviewer #3 stated the following: As for the results of molecular docking, owing to the similar inhibitory activity of rac-6 and DMAT, it will be better for authors to compare the binding modes between rac-6 and DMAT and its deviates with CK2, which will provide a deep insight of binding modes for readers.

RESPONSE 1: We agree with the reviewer's opinion that showing DMAT as a control ligand for both hypothetical complexes with kinases could be valuable for the overall results of the docking studies. Nevertheless, we decided to compare only the most promising inhibitors concerning affinity toward CK2 and PIM1 while also considering their cytotoxicity potential established during the cellular in vivo assays. Moreover, we have arbitrarily decided not to compare the studied compounds directly with the DMAT molecule because this polybrominated derivative is a well-studied inhibitor which crystal structure in complex with protein kinase CK2 is deposited in RCSB Protein Data Bank (PDB code: 1ZOE). For details, please read the paper published by R. Battistutta et al. in Chem. Biol. 2005, 12, 12111219.

Reviewer 4 Report

The manuscript is novel and interesting while a few comments must be addressed to improve the scope of this article.

1. Authors are suggested to trim down the length of the method section and include it into the supplementary part. Same must be followed for the introduction part as it includes unnecessary information.

2. The authors have talked about autophagy but have not shown the results from autophagy marker with the drugs.

3. A simple fluorescence method can be used to calculate interaction of these drugs with both the proteins mentioned CK2 & PIM-1. There is no proof of binding or interaction of proteins with these two drugs in-vitro. Why? The Molecular socking results if considered should be back up by the simulations data as well.

The quality of English could be improved but is optimum for readers to read and follow.

Author Response

Overview of Responses to Reviewer's #4 Comments:

COMMENT 1: Reviewer #4 stated the following: Authors are suggested to trim down the length of the method section and include it into the supplementary part. Same must be followed for the introduction part as it includes unnecessary information.

RESPONSE 1: According to the reviewer's recommendation, some parts of the method section (2.2.1. and 2.2.11) have been transferred from the main manuscript into the Supporting Information document. This action should provide less bulky descriptions in the main text and allow us to avoid self-plagiarism due to recurring procedures. However, in our opinion, most of the information discussed in the Introduction section is necessary for a better understanding of the general topic, especially for readers unfamiliar with this particular field of science. Therefore, we kindly rebut this plea and ask the reviewer to let us leave the Introduction section unchanged. Thanks to this, we hope to have a more in-depth description of the theoretical background of our work and the current state of the art, which should be only beneficial for the overall understanding of this article.

COMMENT 2: Reviewer #4 stated the following: The authors have talked about autophagy but have not shown the results from autophagy marker with the drugs.

RESPONSE 2: We apologize to Reviewer #4 for not using any autophagy marker. We know that the employment of an autophagy marker would have been an excellent solution for this task, but unfortunately, there was no such possibility.

COMMENT 3: Reviewer #4 stated the following: A simple fluorescence method can be used to calculate interaction of these drugs with both the proteins mentioned CK2 & PIM-1. There is no proof of binding or interaction of proteins with these two drugs in-vitro. Why? The Molecular socking results if considered should be back up by the simulations data as well.

RESPONSE 3: In our opinion, the results of the enzyme inhibition studies (please see 3.2.1. Inhibition of Recombinant CK2 and PIM-1), demonstrating an efficient deactivation of recombinant CK2 and PIM-1 kinases in the presence of the developed derivatives, is suitable proof of the existence of the required interactions between novel DMAT-analogs and both studied proteins. Therefore, we would like to kindly rebut this request.

Reviewer 5 Report

The manuscript "Synthesis and Anti-Cancer Activity of Novel Dual Inhibitors of Human Protein Kinases CK2α and PIM-1", written by Winska P, Wielechowska M, Koronkiewicz M and Borowiecki P. describes the synthesis of dual inhibitors of cellular kinases CK2 and PIM1, as derivates of already described dual inhibitor DMAT. Several synthesized derivates were examined for their inhibitory potential toward CK2 and PIM1 in vitro, and for cell cytotoxicity in vivo, on several cell lines. Also, their influence on the induction of apoptosis and autophagy, as well as on cell cycle progression and phosphorylation of target molecules was analyzed. Finally, in silico analysis of inhibitor molecular docking on target molecules is presented.

The manuscript is well written, with comprehensive presentation of chemical synthesis of inhibitors and their analysis. In the Introduction, the roles of CK2 and PIM are presented, as well as chemistry of their inhibitors. Materials and methods are very detailed and comprehensive. Results are well presented. Considering autophagy analysis, it should be stressed that acridine orange staining is not very specific, and it is not clear what the significance is of aggregated or DNA or RNA complexed dye, and what does it mean "yellowish stained lysosomes". In cell cycle analysis, it could be explained whether changes in S phase are statistically significant. In the Discussion, resistance od CML cells could be better explained considering molecular mechanisms; is it possible that CK2 is not inhibited or the cell finds the way to evade the consequences? Also, Bcl-2 cannot be "improved". The role of autophagy in tumor cells is still controversial and autophagy was not unanimously shown in authors' experiments. There are many cellular targets of CK2 and PIM, and each of the cell lines has the overactivation of specific signaling pathways, so wider picture of molecular processes in the cells which include CK2 and PIM could be presented, as well as comparison of inhibitors presented with the effects of other single inhibitors (as at the end of the Discussion).

Other comments:

Line 532: Instead of Scheme 1, it should be Figure (2).

line 600: Vero cells are not "normal"; they are immortalized and have some genetic changes.

Figure 6: explain all the abbreviations.

Author Response

Dear Reviewer #5,

On behalf of all co-authors, I want to sincerely thank you for your time and kind help invested in this revision. We hope that after implementing most of your suggestions, you will find our manuscript much improved and suitable for publication in Pharmaceutics. Below we respond to your comments in point-by-point fashion.

COMMENT 1: Reviewer #5 stated the following: Considering autophagy analysis, it should be stressed that acridine orange staining is not very specific, and it is not clear what the significance is of aggregated or DNA or RNA complexed dye, and what does it mean "yellowish stained lysosomes".

RESPONSE 1: Thank you very much for this suggestion. In order to explain the obtained results in more detail, we have added the following sentence in section 3.2.4: "(…) In addition, OA has the ability to intercalate between the nitrogenous bases of the nucleic acids present in cells and their labeling. DNA or RNA is digested during the autophagy process, and the resulting damage makes it easier to bind with OA molecules. (…)"

The following term: "yellowish stained lysosomes" means lysosomes detected at Ex. 488 nm/Em. 540-550 nm.

COMMENT 2: Reviewer #5 stated the following: In cell cycle analysis, it could be explained whether changes in S phase are statistically significant.

RESPONSE 2: We apologize to Reviewer #5 for not using statistical significance. Therefore, we have corrected Figure 7 and added a statistical analysis of the respective results. Thank you very much for pointing out that issue.

COMMENT 3: Reviewer #5 stated the following: In the Discussion, resistance of CML cells could be better explained considering molecular mechanisms; is it possible that CK2 is not inhibited or the cell finds the way to evade the consequences?

RESPONSE 3: We kindly thank Reviewer #5 for her/his suggestion. We definitely find it beneficial for the overall outcome of these studies. Therefore, additional citations (see Ref: 58 and 59) and the following extract have been added in the discussion:

"(…) Previous studies demonstrated that CK2 expression was increased in leukemia cells from CML patients in blast crisis, as compared to healthy peripheral blood mononuclear cells and showed that Bcr-Abl in K-562 cells physically interacts with CK2α affecting its activity [58]. The therapeutic resistance of K-562 cells can be also partially correlated with an increased metabolic flux towards Warburg phenotype which promotes survival and proliferation [59]. It has been demonstrated that Hexokinase-II (HK-II) is expressed predominantly in cancer cells, which promotes Warburg metabolic phenotype and protects the cancer cells from drug-induced apoptosis. It was proven that K-562 cells have multi-fold high levels of HK-II, glucose uptake and endogenous ROS with respect to normal peripheral blood mononuclear cells [59]. (…)"

COMMENT 4: Reviewer #5 pointed out the following: Also, Bcl-2 cannot be "improved".

RESPONSE 4: The word "improved" was changed to "affect".

COMMENT 5: Reviewer #5 pointed out the following: The role of autophagy in tumor cells is still controversial and autophagy was not unanimously shown in authors' experiments. There are many cellular targets of CK2 and PIM, and each of the cell lines has the overactivation of specific signaling pathways, so wider picture of molecular processes in the cells which include CK2 and PIM could be presented, as well as comparison of inhibitors presented with the effects of other single inhibitors (as at the end of the Discussion).

RESPONSE 5: We agree with the Reviewer's #5 opinion that the role of autophagy is a bit controversial; therefore, we added the additional citation (see Ref: 63) and the following sentence in the discussion section:

"In cancer, autophagy exhibits a contradictory behavior and, depending on the cell type, it may be an important factor for the induction of cell death or tumor progression [63]."

We agree that there are many cellular targets of CK2 and PIM; however, the discussion of signaling pathways in which the protein kinases are involved would be too complex to show in the current manuscript. In our opinion, there is no such possibility in this article to discuss this issue without significant enlargement of the main manuscript's text, which has already contained 72 citations or relevant literature reports. Although we believe Reviewer #5 had a very good and beneficial intention to help us in improving our manuscript, we kindly rebut her/his suggestion, as its implementation would be significantly more appropriate for a review article rather than a full article reporting on experimental research work. Nevertheless, some examples of inhibitor-induced autophagy in K-562 cells have been previously described by us in the discussion:

"Previous studies have demonstrated that a well-established inhibitor of tyrosine kinase, that is imatinib, induces autophagy in the K-562 cell line and in primary cultures of patients with CML [64]. In turn, novel JAK inhibitor ruxolitinib (INCB018424) reported by Lin et al. [65] is able to notably decrease the expression of AKT, mTOR, and STAT autophagy inhibitor genes in K-562 cells, contrariwise control cell line."

We hope that Reviewer #5 will accept our rebuttal in this case and will not change her/his overall positive opinion of our manuscript.

COMMENT 6: Reviewer #5 pointed out the following: Line 532: Instead of Scheme 1, it should be Figure (2).

RESPONSE 6: We take an opportunity to disagree with the Reviewer's #5 opinion as the syntheses pathways are, in general, presented in schemes. In order to give credence to our claim, below we provide an example of the synthesis, which has been reported in one of the recent papers published in Pharmaceutics 2023, 15, 1875 (doi: 10.3390/pharmaceutics15071875). We believe that the shown print screen dispels all doubts concerning this issue:

COMMENT 7: Reviewer #5 pointed out the following: Vero cells are not "normal"; they are immortalized and have some genetic changes.

RESPONSE 7: Of course, we apologize for this misleading. The word "normal" was changed to "non-cancerous".

COMMENT 8: Reviewer #5 asked the following: Figure 6: explain all the abbreviations.

RESPONSE 8: The abbreviations have been explained in the caption as follows:

"(…) Figure 6. Induction of autophagy in K-562 cells. (a) Visualization of intracellular autophagic vacuoles in K-562 cells: fluorescence microscopy of Hoechst (blue-stained nuclei, BF) and AO-stained K-562 cells (green fluorescence ‒ GF and red fluorescence ‒ RF) treated for 48 h with 4 and rac-6 (a fluorescent microscope ECLIPSE Y-TV55, Nikon, x600, magnification), respectively. (…)"

We hope that we have clarified this information.

Respectfully yours,

Dr. Patrycja Wińska

Dr. Paweł Borowiecki

Round 2

Reviewer 4 Report

The author's comments have been answered still the authors should be very cautious while choosing controls in experiments. The article is recommended for publication.

English is good and meet publication requirements.

Reviewer 5 Report

The authors responded to the comments and corrected the manuscript.

English language is fine.

Sentence reconstruction in line 713 is needed.